# Weak ties and the value of social connections for autistic people as revealed during the COVID-19 pandemic
Elizabeth Pellicano [1,2] & Melanie Heyworth [2,3]

A diverse portfolio of social relationships matters for people's wellbeing, including both strong, secure relationships with others ('close ties') and casual interactions with acquaintances and strangers ('weak ties'). Almost all of autism research has focused on Autistic people's close ties with friends, family and intimate partners, resulting in a radically constrained understanding of Autistic sociality. Here, we sought to understand the potential power of weak-tie interactions by drawing on 95 semi-structured interviews with Autistic young people and adults conducted during the COVID-19 pandemic. We analysed the qualitative data using reflexive thematic analysis within an essentialist framework. During the COVID-19 lockdowns, Autistic people deeply missed not only their close personal relationships but also their "incidental social contact" with acquaintances and strangers. These weak-tie interactions appear to serve similar functions for Autistic people as they do for non-autistic people, including promoting wellbeing. These findings have important implications both for future research into Autistic sociality and for the design of practical services and supports to enhance Autistic people's opportunities to flourish.

Social contact is vital for our health and wellbeing[1–3]. Strong, secure relationships with others, or so-called 'close ties', can serve as sources of intimacy and emotional support and can be a powerful antidote to loneliness and social isolation[4]. Despite the importance of close connections with partners, family and friends, there is amassing evidence that interacting with a *diverse* set of social relationships is what matters most to wellbeing[5,6]. This diversity includes close-tie interactions as well as interactions with people on the periphery of our social networks, or so-called 'weak ties'. Although the strength of weak ties has long been recognised in the social sciences[7], these ties have remained strikingly absent from the literature on *Autistic people's* social networks – which has instead concentrated almost exclusively on the importance of close ties[8,9] (see data analysis section of statement on use of identity-first language). Here, we examined the potential power of weak ties for Autistic people, by drawing on data collected during the COVID-19 pandemic – a time during which our access to everyday social practices, including with distant others, was dramatically reduced or, in some cases, non-existent.

## The potential benefits of weak ties

Weak ties have been defined as those relationships that involve less frequent contact – and therefore less emotional intensity and intimacy[10] – than close ties[7]. Weak ties themselves, however, can vary widely. At the very least, they

involve mutual recognition and recurrent interactions, encompassing 'nodding relationships' (such as acknowledging a fellow commuter) or interactions with familiar or 'consequential' strangers (such as the shop assistant at the corner store or dogwalkers in the local park)[11]. But they also include casual acquaintances and more familiar interactions, including with neighbours, co-workers, service providers (therapists, hairdressers), members of a church or synagogue, fellow enthusiasts (chess club, amateur choir, salsa class), or friends of friends (offline or online) – all of whom fall below the threshold for intimate relationships but who nevertheless "pepper daily life"[11].

Granovetter's[7] central thesis was that weak ties connect people to networks that are outside of their usual social circles, helping to diffuse information. In a pioneering study of Boston-based workers, he showed that, while the majority found out about their job through people they knew, most (84%) did so from casual contacts whom they saw only occasionally. Granovetter argued that these weak-tie relationships provided a 'bridge' to different social circles and therefore access to new information, ways of thinking and opportunities – in this case, a new job. This point has been even more definitively demonstrated by Rajkumar et al.[12], who traced the social connections and job outcomes of more than 20 million users on the social network platform, *LinkedIn*, over a 5-year period, testing the extent to which weak ties are causally linked to job mobility. Their findings confirmed the power of weak ties and thus Granovetter's thesis: very weak ties were more

[1]Department of Clinical, Educational and Health Psychology, University College London, London, UK. [2]Macquarie School of Education, Macquarie University, Sydney, NSW, Australia. [3]Reframing Autism, Sydney, NSW, Australia. ✉e-mail: l.pellicano@ucl.ac.uk

robustly related to job changes than strong ties, with moderately weak ties being especially beneficial.

The potential benefits of weak ties are more far-reaching, however. Perhaps the most obvious benefit is that weak ties themselves may develop into close ones, increasing the richness of our immediate social circles. Even for those ties that remain weak indefinitely, they can serve to provide a sense of familiarity and routine to our everyday lives (e.g., saying hello to the security guard each morning at work)[11]; offer social support in times of need (e.g., a neighbour who can collect your child from school at late notice, or virtual community members who share the same experiences)[6]; and be a source of enjoyment with others who share the same interest (knitting club or anime society)[11].

For these reasons, it is not surprising that weak ties have been linked to wellbeing[6,13–15]. Our weak-tie interactions are often considered 'low cost and low stakes', such that they are less emotionally demanding and narrower in nature, often focusing on the activity at hand rather than on the sharing of intimate personal details – and can generate positive experiences. In one study, Sandstrom and Dunn[10] asked people (university students and community members) to track all their social interactions, no matter how minimal, over the course of several different days. Remarkably, weak-tie interactions accounted for 60% of people's daily social contact – and those who interacted with more weak ties experienced greater feelings of belonging and subjective wellbeing, especially on days when they had more weak-tie interactions than usual. Furthermore, the effect of weak-tie interactions on belonging was particularly strong for people low in extraversion, suggesting this group might especially benefit from weak-tie interactions.

Moreover, social networks that are diverse in nature – consisting of a range of weak ties *and* close ties – are more beneficial than networks comprised mainly of close family or friends. Such relational diversity has been linked to positive outcomes, including lower risk of mortality[16], protective effects on physical health[17], reduced loneliness[18], enhanced physical activity and positive mood[19], as well as greater subjective wellbeing[5,10].

Despite the apparent benefits of relational diversity, and of weak-tie interactions in particular, almost all the research on Autistic people's social interactions has focused on close ties with friends and family and, to a lesser extent, intimate partners[20,21]. This emphasis is probably due to the long-standing view that Autistic people do not value social relationships in the way that non-autistic people do. Difficulties making and keeping friends have long been considered 'core' features of an autism diagnosis[22] and dominant explanations have suggested that Autistic people lack the motivation[23] and/or cognitive skills[24] for social interaction, which prevents them from establishing and maintaining the types of reciprocal relationships important for wellbeing. Consequently, many interventions have focused on improving Autistic children, young people and adults' social competence, and thus their capacity to build and maintain meaningful friendships, by targeting sociocognitive 'deficits' early during development (e.g., joint attention[25]; social rewards[26];) or providing formal social skills training in adolescence and adulthood[27], see[28] for discussion.

Recent critiques, however, have suggested that this narrow focus on Autistic people's social challenges has resulted in a radically constrained understanding of Autistic sociality[29,30]. Indeed, research prioritising Autistic perspectives has instead demonstrated that many Autistic people *want* to connect with friends, family and lovers[31,32], even if negotiating those relationships can be challenging[33–35], and that such connections, especially with other Autistic people[36,37], are important for Autistic people's wellbeing[30,38] and sense of belonging[39]. Indeed, *not* securing the kinds of close ties that Autistic people desire can lead to loneliness[40,41], which is one key predictor of poor mental health[42], self-harm[43] and suicidality[44,45].

This recent work suggests that social relationships are essential to good Autistic lives, and therefore gives us some reason to expect that Autistic people might appreciate – and benefit from – a diversity of social interactions, including weak ties. A plethora of research on Autistic people's experiences in schools, workplaces and clinical settings already provides some (indirect) evidence to support this possibility: Autistic people's experiences are better when they have positive interactions with their classmates and teachers[46,47],

their co-workers[48] and professionals supporting them[49,50] – people considered to be on the periphery of their social circles, or weak ties.

One study has *directly* investigated the breadth of Autistic people's social interactions and experiences, ranging from weak-tie encounters with familiar strangers and known acquaintances to close friends. Chan et al.'s[51] qualitative analysis examined Autistic adults' social experiences across multiple contexts, including workplaces and education settings, neighbourhoods, common interest groups (e.g., games nights, faith-based communities), support services and inclusive environments, and online networks. Autistic adults reported these experiences to be meaningful and made them feel part of their communities, especially when the spaces (online or offline) offered a sense of acceptance and safety. Chan et al.'s[51] findings demonstrated that understandings of Autistic sociality cannot be limited to close friendships and intimate relationships, but need to consider the full range of social experiences, across a range of (weak- and close-tie) interactions and contexts.

Chan et al.'s[51] initial findings suggest that weak-tie interactions can be beneficial for Autistic people. Here, we investigated this possibility further, utilising data from a broader study on Autistic people's experiences of the COVID-19 pandemic[32]. The pandemic and its associated restrictions caused severe disruption to our everyday lives, preventing access to our usual routines and social practices – in workplaces, playgrounds, schools, grocery stores and places of worship. The 'stay-at-home' orders meant that we interacted with fewer people on a daily basis, at least offline, thereby shrinking our social circles, especially our weak-tie relationships[14]. Far from welcoming the isolation that came from lockdown, previous work showed that our Autistic participants intensely missed seeing their friends and family in the same way as everyone else – and also suggested they even missed fleeting, incidental interactions[32].

In the current study, we explore in greater depth Autistic people's discussions of their weak-tie interactions in the context of pandemic-related restrictions. Specifically, we asked, what did the COVID-19 lockdown reveal about the potential power of weak ties for Autistic people?

## Methods
### Participants
We drew on a rich set of interviews conducted during the initial stage of the COVID-19 pandemic (May – June 2020). Here, we focus specifically on the 95 Autistic participants from that study, including 16 young Autistic people and 79 Autistic adults, drawn from a convenience sample, recruited through social media. To be eligible, participants needed to be English speaking, and willing and able to convey their experiences of the pandemic. Most participants came from Australia ($n = 83$; 87%), where the restrictions used to curb the COVID-19 virus were particularly severe, and reported themselves to be predominantly of white ethnic background and moderate-to-high socioeconomic status. Most participants ($n = 70$; 74%) reported staying at home following government physical distancing rules, while the remaining 25 (26%) were self-isolating either due to suspected COVID-19 ($n = 2$; 2%) or another health vulnerability ($n = 23$; 24%).

Our young person participants included five girls, eight boys, two with non-binary gender and one who was gender-questioning, aged between 12 and 18 years (Table 1). All had received a clinical diagnosis of an autism spectrum condition, on average, at the age of 8 years 4 months (range = 2–16 years).

Our adult participants included 61 women, 12 men, four people identifying as non-binary and two as 'other', aged between 23 and 69 years. Most adults reported having received a clinical diagnosis of autism ($n = 67$; 85%), largely during adulthood, while the remaining self-identified as Autistic ($n = 12$; 15%). Thirty-five (44%) also reported to be parents of Autistic children. Prior to the COVID-19 pandemic, 46 adults (58%) were in some form of employment (Table 1). One fifth of our sample ($n = 15$, 19%) had experienced a change in their occupational status during the COVID-19 outbreak.

Most young people ($n = 12$; 75%) and adults ($n = 70$; 89%) reported co-occurring diagnoses of often-multiple neurodevelopmental and/or

## Table 1 | Participant characteristics

| | Autistic adults (n = 79) Mean (SD) Range or N (%) | Young people (n = 16) |
|---|---|---|
| Age (years) | 40.61 (9.64) 23.29–69.35 | 14.95 (2.08) 12.14–18.41 |
| Age at autism diagnosis (years)[a] | 35.21 (12.32) 2–64 | 8.38 (3.72) 2–16 |
| Interview duration (min) | 54.04 (20.72) 15.20–99.44 | 28.81 (11.02) 19.20–50.40 |
| Gender | | |
| Woman/girl[b] | 61 (77%) | 5 (31%) |
| Man/boy[c] | 12 (15%) | 8 (50%) |
| Non-binary | 4 (5%) | 2 (12%) |
| Other | 2 (2%) | 1 (6%) |
| Country of residence | | |
| Australia | 68 (86%) | 15 (94%) |
| New Zealand | 1 (1%) | 0 |
| Philippines | 1 (1%) | 0 |
| UK | 8 (10%) | 1 (6%) |
| USA | 1 (1%) | 0 |
| Predominant racial/ethnic background | | |
| Australian Aboriginal | 1 (1%) | 0 |
| Chinese | 1 (1%) | 1 (6%) |
| East Asian | 0 | 0 |
| Mixed | 5 (6%) | 0 |
| White Australian/New Zealand | 14 (18%) | 1 (6%) |
| White European | 54 (68%) | 12 (75%) |
| White Other | 1 (1%) | 2 (12%) |
| Prefer not to say | 3 (5%) | 0 |
| Living arrangements | | |
| Alone | 18 (23%) | – |
| With partner only | 14 (18%) | – |
| With partner & children | 27 (34%) | – |
| With children only | 7 (9%) | – |
| With relatives | 9 (11%) | – |
| With friends | 4 (5%) | – |
| Highest qualification | | |
| Completed primary school | 1 (1%) | – |
| Completed Year 10 | 4 (5%) | – |
| Completed high school | 10 (13%) | – |
| Vocational training | 15 (19%) | – |
| Undergraduate degree | 22 (28%) | – |
| Postgraduate degree | 27 (34%) | – |
| Pre-COVID-19 occupational status | | |
| Part-time employment | 15 (19%) | – |
| Full-time employment | 19 (24%) | – |
| Self-employed | 12 (15%) | – |
| Studying | 10 (13%) | – |
| Full-time parent | 13 (16%) | – |
| Unable to work due to disability | 4 (5%) | – |
| Unemployed & seeking work | 5 (6%) | – |
| Retired | 1 (1%) | – |
| Co-occurring conditions[d] | | |
| ADHD | 28 (35%) | 5 (31%) |

## Table 1 (continued) | Participant characteristics

| | Autistic adults (n = 79) Mean (SD) Range or N (%) | Young people (n = 16) |
|---|---|---|
| Anxiety disorders | 43 (54%) | 9 (56%) |
| Autoimmune disorders | 11 (14%) | 0 |
| Bipolar disorder | 6 (8%) | – |
| Chronic fatigue syndrome | 3 (4%) | 0 |
| Chronic pain | 15 (19%) | 0 |
| Depression | 49 (62%) | 5 (31%) |
| Drug/alcohol dependence | 2 (2%) | – |
| Dyslexia | 4 (5%) | 1 (6%) |
| Dyspraxia | 1 (1%) | 0 |
| Eating disorders | 9 (11%) | 2 (12%) |
| Epilepsy | 1 (1%) | 0 |
| Gastrointestinal issues | 22 (28%) | 0 |
| Intellectual Disability | 2 (2%) | 0 |
| OCD | 6 (8%) | 1 (6%) |
| Personality disorders | 5 (6%) | – |
| PTSD | 22 (28%) | 0 |
| Schizophrenia disorders | 1 (1%) | – |
| Sleep disorders | 17 (22%) | 4 (25%) |

Data are mean (SD; range) or n (%). Percentages may not sum to 100% due to rounding issues.
[a]n = 67 (remaining participants self-identified as autistic); [b]Included transgender women; [c]Included transgender men; [d]Participants could select all options that applied to them. Percentages therefore do not add to 100.

psychiatric conditions, including especially anxiety (n = 52; 66%) and/or depression (n = 54; 68%) (Table 1).

### Procedure
Ethical approval for this study was granted by the Human Research Ethics Committee at Macquarie University (Project ID 6665). All interviewees provided written informed consent prior to taking part; parents provided additional consent for young people under the age of 16 years. We followed the Standards for Reporting Qualitative Research guidelines (see Supplementary Table 1)[52]. This study was not pre-registered.

To begin, participants, or in the case of young people, participants' parents, completed an online questionnaire to provide information about demographics, clinical diagnoses and other background details, including COVID-related information and isolation status. All young people, and especially those aged 16 years and over who consented for themselves, were given the choice to complete the background questionnaire or for their parent to do it on their behalf. Only one of the young people agreed to complete the online questionnaire themselves. We elicited participants' experiences of the COVID-19 pandemic through semi-structured interviews, conducted via their preferred means of communication, including video-conferencing (Zoom) (n = 73; 77%), telephone (n = 8; 8%), email (n = 12; 13%), or live text-based chat (n = 2; 2%). Interview questions centred on participants' perceived impact of the pandemic on their living, working and learning arrangements, social relationships, access to services and sense of wellbeing (see Supplementary Table 2). Participants received the primary interview questions ahead of the interview. Spoken interviews varied in length, ranging between 15.20 and 99.44 mins (Table 1). All participants received an AUD25 (or equivalent) voucher as a thank you for their time.

### Data analysis
All spoken interviews were recorded with participants' prior permission and transcribed verbatim. The resulting, de-identified transcripts were sent to participants for checking prior to analysis. We followed Braun and

**Fig. 1 | Thematic map.** Figure illustrates identified themes and subthemes on Autistic people's experiences of weak-tie interactions in the context of the COVID-19 pandemic.

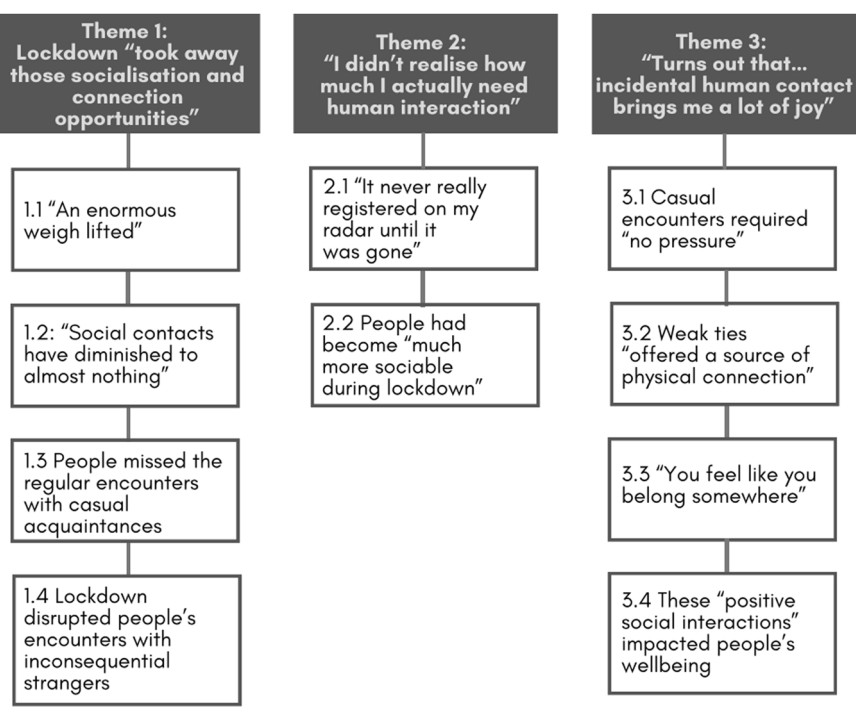

Clarke's[53] method for reflexive thematic analysis within an essentialist framework, in which our goal was to report the meanings and experienced reality of the participants – in this case, about their peripheral social relationships within the context of the pandemic restrictions. We used an inductive approach to identify patterned meanings within the dataset. We sought to stay close to participants' language and therefore also adopted a semantic approach. One researcher (EP) re-read all transcripts, taking reflexive notes and applying codes to those parts of the transcripts deemed relevant to the research question (managed in NVivo, version 12). The same codes were then applied to all transcripts. Following discussion with MH, codes were clustered together to identify candidate themes and subthemes. EP then generated a draft thematic map, and the relevant data were collated under each theme and subtheme. The draft analysis was reviewed and revised by MH and then discussed and revised multiple times. The themes were also discussed with some participants ($n = 5$). Analysis was therefore iterative and reflexive[53].

**Researcher positionality.** EP is a non-autistic autism researcher and psychologist, committed both to ensuring that autism science reflects the realities of Autistic people's lives and to conduct such research in partnership with Autistic people and their families and allies. MH is an Autistic autism researcher and an advocate working for and with the Autistic and autism community to build Autistic wellbeing and dignity through education and co-produced research. We believe in the benefits of the social model of disability and a neurodiversity approach to autism research and practice. These beliefs and assumptions will have influenced all aspects of the research, especially the analytic process. It also influenced the language we use throughout this paper. We use identity-first language ("Autistic person") because it is often preferred to and considered less stigmatizing than person-first language ("person with autism")[54–56], and is preferred by MH. We also capitalise the word Autistic to reflect its status as a valued identity and shared community.

**Community Involvement**
No lay community members were involved in this study. The original study was co-produced by a team of Autistic and non-autistic researchers, who were actively involved throughout the research process[32]. For the current study, one Autistic and one non-autistic researcher from the original team

analysed the existing data to address the current research question. Working together in this way resulted in collaborative decisions on the research question, the analysis and interpretation of the data and the write-up of the results. It also meant that the results were interpreted through a strengths-based, rather than deficits-based, lens.

**Reporting summary**
Further information on research design is available in the Nature Portfolio Reporting Summary linked to this article.

## Results
We identified three themes that addressed our research question, including, Lockdown "took away those socialisation and connection opportunities" (Theme 1); "I didn't realise how much I actually need human interaction" (Theme 2) and "Turns out that… incidental human contact brings me a lot of joy" (Theme 3) (see Fig. 1). Themes and associated subthemes are also numbered below and presented in bold and italics, respectively. Illustrative quotes are attributed via participant IDs.

### Theme 1: Lockdown "took away those socialisation and connection opportunities"
Initially, many people described the isolation caused by the COVID-19 lockdown as a blessed relief – *"an enormous weight lifted" (subtheme 1.1)* (210Adult). Lockdown appeared to free them from some of the overwhelming demands and expectations of living in a neurotypical world, including from "not having to people… to do all the usual small talk and loads of masking, which is just really exhausting. And all the appointments because we're all quite complicated which are, again, massively anxiety-provoking. All that sort of stress" (234Adult). One young person elaborated:

> People-ing takes up energy to have a conversation, to be normal. And that's hard for me. Think of it like running. I don't want to run perpetually, so I have to have a break at some point. And people-ing is hard and I'm not a marathon runner. (109YP).

"Not having to interact with people a lot" (102YP) also meant that they did not need "to be in situations where I would experience anxiety"

(031Adult), especially in ones over which they felt they had less control, like at work where "if I go into the kitchen, there might be a person there that I need to summon some social skills to have a conversation" (019Adult), or at school "with the school mums, you get shut out and excluded because you're just that little bit different, or a lot different maybe" (222Adult). Instead, without the "pressure to see people" (036Adult), many people felt they could "just be ourselves" (211Adult), which "positively affected my mental health" (025Adult).

This initial relief, however, did not last. Instead, people emphasised that the social implications of the lockdown restrictions were more far-reaching and "far more isolating" (216Adult) than they had anticipated: "I didn't realise how much the restricting would impact us" (235Adult). "Being locked away" (025Adult) impacted people's more intense personal relationships, preventing them from "seeing my friends" (109YP) and family. But they also spoke of how it dramatically affected their broader interactions with casual acquaintances, co-workers, classmates and strangers: "Without realising it, *social contacts have diminished to almost nothing*" (*subtheme 1.2;* 032Adult).

Interviewees described how access to many places and the practices they enabled was often curtailed. They told us that they "don't get to the places I usually go – the library and places like that" (110YP), or "big shopping centres" (107YP), and that their children could no longer "go to the playground or go to the shops" (209Adult), or "have his favourite pizza from his favourite shop" (230Adult). Adults, too, missed "going to the pub and watching the football" (042Adult), "playing table tennis" (234Adult), and going "to an art gallery or museum" (013Adult). One adult summed it up: "pretty much everything in my life that I loved doing was just gone" (043Adult). While people did not necessarily explicitly speak of these activities as opportunities to socialise, a unifying feature of them was their inherently social potential, replete with the prospect of casual, 'weak' interactions. Thus, interviewees identified that "you start to miss out" (116YP) from "not being in the same space as people and things" (005Adult).

The now-restricted access to their usual places and practices meant that *people missed the regular encounters with casual acquaintances (subtheme 1.3)*, including in their local community, with sports clubs ("usually my social interaction is at training with my team") or at "my favourite coffee shop – they had my keep-cup there for the whole of COVID and they saw me walk past and called out 'your keep-cup's here and we filled it with coffee for you' and that was really nice" (044Adult). One man explained the emotional impact of this disruption:

> Before COVID, most Fridays, I would go down to the aged care place where my grandma was and enjoy the music concert there… I'd often talk to people in the lounge room for five, ten minutes and then I'd take grandma up to her room and help her get ready for bed. When grandma died, the staff told me that I was welcome to keep coming to the concerts… One thing that affected me was the whole not being able to visit the care home anymore. I'd been visiting there regularly in the 23 months since grandma had died and I felt like that rug had just been pulled out from underneath me. (015Adult).

Interviewees also missed their regular interactions at work, which "totally changed when we had to work remotely… my social life was really built around my job" (037Adult). Others agreed that "the only time I'd get to actually see other people was when going to work" (020Adult) and, as such, they "really missed the in-person stuff" (235Adult), even when getting to work was less-than-straightforward:

> The biggest thing is I miss getting the train. I hate getting the train to and from work because it's an hour-and-a-half each way and peak hour just gets so packed, but I just miss the routines I had in place and I miss the going into work, seeing people. (005Adult).

Young people also felt the social impact of learning from home. They reported being "frustrated and sad and angry… that I don't get to choose when I socialise with someone" (103YP). They reported missing their classmates at school, even the ones "who are just always messing about and getting on my nerves. But now I'm almost longing to just see them running down with a load of crisp packets wanting to sell them to each other" (116YP).

"Being trapped at home for such a long time" (103YP) also *disrupted their encounters with seemingly inconsequential strangers (subtheme 1.4)*, not just casual contacts. People talked about how they missed "not being able to fuss over other people's dogs, because you do that when you're a dog owner – when you see people, you do stop and say hello, because it's easy to talk about pets. It's a positive social interaction" (007Adult). People also described the pleasure of other fleeting, face-to-face interactions while "I'm travelling – I like meeting people on the train or at the train station" (208Adult), or when out-and-about for work:

> I have a small franchise, which is all in lockdown at the moment, and I've got various vending machines dotted around, and I go along, I take the money out to pay in the bank, I refill the vending machines [with sweets] and say hi to people where I go. That's mostly my social contact. (035Adult).

## Theme 2: "I didn't realise how much I actually need human interaction"

Our interviewees repeatedly reported being surprised at how much they missed broader social connections. Adults spoke of how "we're not big social people" (235Adult), but had come to notice "how much we did miss the broader social group", because their social circles had dwindled to the extent that "you were having the same conversations with the same people" (223Adult).

This was especially prominent for participants for whom incidental social contact was their "primary adult interaction for the day, because I'm a stay-at-home parent" (229Adult). They recounted how they had come to "realise how important that incidental human contact was to me… it was so incidental that *it never really registered on my radar until it was gone*" (subtheme 2.1) (008Adult). This same interviewee elaborated:

> The paying for the petrol at the service station, as much as it shits me, because I work with an assistance dog right, so everywhere I go, I've got this giant, white, majestic beast next to me. So, the soundtrack of my life is, 'oh my god, it's a dog. It's a beautiful dog. What kind is he? How old is he? What does he do?' And, like, that drives me bananas most of the time. And I've actually found myself even missing that. [008Adult].

Others, too, noted how "I thought I was fine without people… but this has taught me that, actually, I do need people" (234Adult): "that I am a social creature to some degree was a revelation to me" (225Adult). Young people shared similar experiences:

> Before, I just took them for granted, as they're just the irritating person at the back of the classroom, and then I've realised how much I would actually miss them. Mind you it isn't a lot, but I wasn't expecting to miss them at all. (101YP).

To compensate for the loss of these broader connections, some reported they had become "*much more sociable during lockdown*"

*(subtheme 2.2)* (216Adult) – they were "tolerating idle small talk with random people a lot more" (214Adult) and "needing a lot more contact from my friends than I perhaps usually would" (008Adult). People also reported seeking out the casual encounters they were missing in their local community, having "more connections with the neighbours than usual" (039Adult), and "making more of a real conversation than the obligatory 'hi, how are you, I'm fine, thanks, move on'. I actually met nice people down the street that I had never met before. I think I appreciated those things" (233Adult). Others went further and developed new routines for social connection:

I've probably been more chatty than normal… I've got a café on the corner of my street which has stayed open just for takeaways – I actually started going there pretty much every day. There's a few people who are always there – from school and from scouts. I think I was more willing to do the chit-chat thing because I knew I wasn't going to be stuck there. (018Adult).

As one interviewee put it: "I'm hoping that this has taught us the value of connection" (008Adult).

### Theme 3: "Turns out that… incidental human contact brings me a lot of joy"

Our interviewees also spoke of why the loss of these broader social connections seemed to matter so much. They emphasised that the casual encounters with people at work, at the school gate, and in the community required *"no pressure" (subtheme 3.1)*. Some felt comfortable in these interactions because they were familiar people in familiar contexts:

I thought I was a really autonomous blob living on my own island and turns out that that sort of incidental social interactions brings me a lot of joy. I think the reason, and this is just speculation, is because there's not very much pressure on it. I've been working in the same place for close to five years now and I know these people super well and it's not difficult, so it feels really nice to have those incidental moments that aren't taxing on me. (225Adult).

People also reported they valued interactions with casual connections, especially those that were structured in some way, often around hobbies or passions, like "crafting club… it's really just chill. You're just sitting there crocheting, there's no pressure to talk, and people are talking about this, that and the other, and you can chime in" (204Adult). Many spoke of how they "suck at small talk" (218Adult) but felt that "the style of socialising alongside activity really works for me" (002Adult): "I play Scrabble with a very eccentric group, lol. I like that because Scrabble gives us a topic to talk about and I do not have to think up small talk" (032Adult). One person explained how the context can facilitate interactions with strangers:

The local park across the road, it's suddenly got quite busy with people walking their dog. So, I would go and regularly see the same people. And after a while, you walk past them and make eye contact or say hello. And I've really enjoyed that. I realised I don't mind that recognition of another local community member, recognising each other, saying hello. I hate small talk, but around dogs it's really easy. It's the same script that you just run every time. (203Adult).

These examples also highlighted the sense of agency people could have about when and how they interacted with casual contacts/strangers, namely "where I can control when they happen" (005Adult). The "incidental five-to-ten-minute chats" (221Adult) were the interactions they could choose to have:

On Thursdays, I would take my daughter to ballet class, and the mums would go to buy a coffee and chat… It's just this nice one hour of socialising – I'm not going to have to make an excuse to leave – and then we all go our separate ways and get on with our days. I miss that. (209Adult).

Interviewees noted that, in some cases, the regular face-to-face interactions with casual contacts also *offered a source of physical connection (subtheme 3.2)* that they rarely got the opportunity to enjoy, and that lockdown took away: "the 'not being able to hug people' thing" (024Adult). They spoke of how they missed their expected interactions with people at, for example, their weekly Bible Study group, where "sometimes we hug each other" (033Adult), or at their workplace: "they [charity volunteers] used to actually give us hugs… and it's the only form of physical interaction that I get. I really miss that" (007Adult). Another recounted how they longed for the physicality of these interactions:

I used to do circus classes most nights and weekly dnd games. It was really hard to stop doing those, both for the activities themselves (special interests and sensory needs) and because they're my main source of direct contact with people. I would always hug these people, so it's been really hard not having hugs with them too. (002Adult).

People's responses also illustrated how their interactions with acquaintances and strangers imparted a sense of social belonging: *"you feel like you belong somewhere" (subtheme 3.3)*. They felt this connection through their "big Autie passions": "my choir has been a big part of helping me feel like I am the person who enjoys being with others and all that… they showed me some of the different ways of being autistic" (017Adult). This feeling was also engendered in the workplace, where people reported that they liked "the security of having those people around you" (024Adult). One person elaborated:

Actually, I've realised that I rely very heavily on my workplace relationships because I don't have that many friends and relationships outside my family and work. I realised that the interaction that I have in the office is actually quite important to me, so that was a challenge. I have a deep sense of belonging there. (225Adult).

They also reported how the restrictions affected their sense of social participation. They reportedly "missed having people" (011Adult) around them, "that community and lack of normality" (013Adult). One person noted they were "really good at making acquaintances – I'm not very good at making it go beyond that. But I still miss having all those acquaintances, just having people around" (204Adult). They "really wanted life to go back to normal as much as possible" (229Adult) so that they could "mix with everyone like we normally did" (042Adult), to be able to have the choice "to sit by myself in a café and be surrounded by people" (222Adult). In this way – and even in the absence of direct social interaction – "I know I'm not alone in the world" (038Adult).

Perhaps for these reasons, it is unsurprising that interviewees also conveyed how much *these "positive social interactions" impacted their wellbeing (subtheme 3.4)*. Some described how "I get my energy from working with people and working with the kids [at school] particularly" (017Adult), and that "just by seeing someone else that you like being in contact with makes it feel less lonely" (007Adult). They reported how the absence of these broader social connections had a detrimental effect on their mental health, making them feel "more needy and anxious, and needing a lot more contact from my friends that I perhaps usually would" (008Adult):

The lack of incidental contact with colleagues really had a bit of an impact on me. Living alone, having those interactions with my students and colleagues is pretty much enough for me. And then I go home and have some solitude and recoup and go back into it the next day. But without any of that I found that quite difficult. (209Adult).

Interviewees explained further how this "incidental human contact" was fundamental to their wellbeing. One person spoke of the boost they would get seeing "that guy that wears his headphones every day and we just wave at each other. So, yes, that has really improved my wellbeing, and that surprised me" (203Adult). They were adamant that "it's good for me to have some community contact aside from family" (032Adult) and that "it's really healthy mentally to be able to connect with other people and what they're doing. Having those outside interactions is positive and healthy" (035Adult).

## Discussion

Examination of the diversity of Autistic people's social connections, especially the extent and nature of their weak-tie interactions, has been virtually absent from the academic literature. Here, we examined this topic in the context of the pandemic restrictions of 2020. Contrary to stereotypical assumptions, the vast majority of Autistic people we spoke to reported deeply missing the social contact that used to be a key aspect of their world, even when they may have initially welcomed the isolation that came from lockdown. Lockdowns most obviously affected the more intense personal relationships. Our in-depth analyses here clearly demonstrates, however, that this loss was also more far-reaching, affecting Autistic participants' "incidental social contact", including their interactions with casual acquaintances and strangers. Moreover, this loss appears to have a significant, negative effect on their wellbeing.

This reduction in Autistic people's weak-tie interactions during the pandemic is consistent with studies conducted with general population samples. For example, in a large German sample, Bertogg and Koos[57] found that many people experienced a significant shrinkage in their social networks as a result of the COVID-19-related stay-at-home orders, losing both acquaintances and friends – with larger effects on people's weak (rather than close) ties, and in some groups in particular (e.g., young people, migrants). Another study demonstrated similar findings in the context of the workplace, where weak ties are likely to form from chance encounters at the office. Carmody et al[58]. showed that mandatory working-from-home significantly hindered the formation of weak ties for US-based workers and weakened the spread of information among co-workers. It is usually assumed that Autistic people's social circles are often smaller than those of non-autistic people – sometimes by design, sometimes not – consistently mainly of close relationships ('strong ties'). It is plausible, therefore, that the loss of weak-tie interactions during lockdown had a disproportionate impact on Autistic people in particular, shrinking their social networks further. It is also plausible that the curtailing of weak-tie interactions affected Autistic people's social networks in similar ways to the general population and similarly disproportionately.

It is hard to be certain of this, of course, given that we know so little about the diversity of Autistic people's social networks and the extent and nature of their weak-tie interactions. We contend that the relative absence of such research is one example of 'undone science'[59] – that is, cases of research areas not being pursued because the neuronormative lens of the prevailing paradigm leads scientists to characterise some areas as not worth studying. Indeed, conventional understanding of Autistic people's sociality has been dominated by research designed to understand the underpinnings of their 'social deficits', with prominent theories reinforcing long-held ideas that Autistic people cannot or do not want to form social relationships[23,60]. Our findings resoundingly contradict the foundational premise of these influential theoretical accounts, by showing that Autistic people want to connect with others, including with casual contacts and even strangers. The over-reliance on individualistic explanations of Autistic 'social deficits', together with failing to take Autistic testimony seriously[29], are likely to have precluded understandings of Autistic sociality, especially the diversity of people's social circles.

In particular, these conventional explanations have overlooked the contextual nature of Autistic sociality. Our Autistic participants – young people and adults – spoke of the reduction in their weak-tie interactions across a range of different contexts. People spoke of how they missed not being able to go to the library, to shopping centres and to art galleries – places that are inherently social in nature and afford opportunities to interact with inconsequential strangers or which require simply being around people in order to be fully enjoyed. People also spoke of the importance of the places and practices that provided a coming together through shared interests, like Dungeons and Dragons or choir clubs. Such structured, interest-focused social interactions are often enjoyable because they are low stakes and low intensity, with social interactions focused on the activity or interest, rather than on the individual[61,62]. Losing access to these kinds of interactions resulted in losing access to opportunities to connect with others in casual but highly motivating and meaningful contexts.

In contrast to our view, some have suggested that the link between engaging in shared interest groups and subjective wellbeing exists only because it provides a context for Autistic people to develop sustained, meaningful relationships[63]. Our own findings, however, reinforce those of others[36], which suggest that the effects on wellbeing may arise from fleeting interest-based exchanges with others, as well as sustained ones. In analysing the reports of young Autistic adults working within an Autistic workspace, Bertilsdotter Rosqvist[36] revealed how much they valued interactions with others that are driven by genuine shared interests, which they felt could be a means of making contact with other people, however casually, and helped to confer a sense of belonging within a community. Bertilsdotter Rosqvist[36] further proposed that these interest-driven social interactions are facilitated by being within Autistic-dominated spaces. Our findings suggest, intriguingly, that such interactions may also occur with non-autistic people – and that it was the loss of these often short-in-duration, but interest-related interactions during the COVID-19 pandemic that was also unsettling for our participants.

People also lamented not being able to access their expected, routine interactions, such as the park where they walked their dog and interactions with other dog-walkers, the local café where they could chat casually to the barista, their workplaces where they mingled with colleagues, or the school gate where they engaged with other parents. It was these apparently casual interactions in neighbourhoods, or workplaces, or community settings, which our participants spoke of having embedded into their regular, pre-pandemic everyday lives and which offered comforting and connected social predictability. They reported having previously taken them wholly for granted – or not even noticed their importance – and yet when they were taken away, their absence made a significantly detrimental difference to their wellbeing and sense of belonging in the world. It was not just the interruption to routine *per se* that was raised here – as might be expected – but the interruption to a particular kind of social routine.

This was illustrated acutely in people's reports of how much they missed not being able to physically spend time with their colleagues at work. Workplaces are typically understood to be particularly challenging and uncomfortable for Autistic people, involving often-unpredictable daily interactions with colleagues, whether of the kind requiring small talk around the 'water cooler' or equivalent, or the more transactional form of interaction often described as "networking to progress"[64]. Here, however, the people we spoke to described workplaces as important conduits to social fulfilment, not just social overwhelm, especially when it was possible to choose with whom, where and how to make these social, collegial interactions. As one of our participants attested: "I don't like that the choice is taken away". Our findings in this regard echo those of Chan et al.[51], whose participants also spoke of how employment settings were a place in which they experienced positive interactions with others and could offer a sense of belonging.

Drawing all of this together, our data critically illustrate how Autistic people's fulfilling social relationships are inseparable from their everyday

practices and places. This is why "weak ties" cannot be overlooked. For Autistic people, as well as their non-autistic peers, "weak ties" ultimately foster a sense of belonging and are critical to wellbeing[6,13–15]. Such findings have potentially profound implications for both ongoing research and broader practice. When taken seriously, it is clear that creating or enabling inclusive and accessible places for Autistic people casually to meet with people from beyond their own personal circles is a vital task.

This should lead Autism researchers to consider the insights of those who have written extensively about the means by which we can shape spaces that support "bridging" relationships between acquaintances and strangers. Ever since Jane Jacobs' seminal *The Death and Life of Great American Cities*[65], researchers examining the impact of the built environment on neurotypical social interactions have argued that "bumping spaces" (unintentional encounters) and "gathering places" (for intentional places to meet) are intimately connected to living well[65,66]. But these insights have until now been overlooked in the autism literature, which focuses almost exclusively on Autistic people's own proclivity – or supposed lack thereof – for community participation. Our findings suggest that autism researchers should turn their attention to the form of places "where the casual or informal encounters occur during the daily routine of life", including the social composition of different neighbourhoods and the nature of the natural and built environment, all of which play a part in forming and maintaining social interactions that provide opportunity for enhancing wellbeing[67].

There are further reasons to believe that facilitating everyday weak-tie interactions might be particularly advantageous at certain key moments in Autistic people's lives. It is well noted in the non-autistic population that outer social circles (our 'weak ties') tend to shrink compared with our inner ones (our 'strong ties') as people get older[68]. It is also widely acknowledged that, in older adults, having a larger number of weak ties is longitudinally predictive of both lower depressed affect and higher positive affect, suggesting that weak ties might promote emotional wellbeing over time[69]. Encouraging connections with strangers and acquaintances in older age is thus a key part of many clinical and social interventions in non-autistic people[70,71], and our work here suggests it may be relevant in Autistic communities, too. Similarly, among neurotypical young people, it is recognised that vital social supports often follow from weak rather than stronger ties, especially during emerging adulthood[72]. Weak ties often lead to introductions to people from beyond an immediate family or a community, for example, in ways that might support progress in key areas of life, including finding a job, seeking care for oneself or one's dependents, or connecting to new hobbies or pass-times[73]. It is worth considering whether these patterns could extend to Autistic young people too, and to consider what would need to be put in place to ensure that they work effectively.

That said, it is important also to remember that Autistic people's weak-tie interactions may still be critically different from non-autistic people's, and efforts to promote them should be attentive to those differences too. Those to whom we spoke did reveal anxiety about certain kinds of casual interactions – especially where they had little choice but to engage in them or when "doing well" in an interaction appeared to be directly consequential for other aspects of life, such as leading to a workplace promotion. In making any suggestions for the future, it will be crucial to more deeply understand and attend to these experiences. It is sometimes suggested in the literature on social interaction among neurotypical people, for example, that people should be "nudged" into greater social interaction even against their original desire to avoid such interaction[74]. In practice, this can result in policies which force socialisation in prescribed ways and for specific purposes, but there remain strong reasons to suppose that such interventions would not be conducive to Autistic people's wellbeing and should largely be avoided.

It is also important to recall that not all weak-tie interactions will be positive for Autistic people, especially when social stereotyping, stigma and bullying remain commonplace. There is a plethora of research demonstrating negative interactions with people who might be on the periphery of Autistic people's social circles, including school peers and teachers[46,47], co-workers[48] and health-care professionals[49,50] – and might serve to diminish Autistic people's sense of belonging and experiences of social exclusion.

Understanding the diversity of these weak-tie interactions – the positive and the negative – will be an important next step for research. Facilitating interactions where Autistic people feel as if they have some element of control are welcomed, perhaps through the structured socialising of a shared interest group or the interactions of predictable, routine engagements, and that give them the boost that our participants spoke of, will be crucial to their flourishing.

## Limitations

This study has several limitations. First, these findings are not fully representative of the Autistic population. Our sample was well educated, largely from white racial/ethnic backgrounds and the gender distribution – at least for our Autistic *adult* sample, which included many more women than men – does not reflect population norms for diagnosis (as more men receive a diagnosis[75]). Our unusually high sampling of women is likely owing to our inclusion of Autistic parents, who made up almost half (44%) of our sample of Autistic adults, and were mostly mothers (94%), who also reported on their Autistic children[32]. Although these potential sampling biases are not unusual for online studies like this one[56,76,77], they are a perennial problem in autism research[78], just as they are in broader psychological research[79]. Our sampling and interview methods – prompted by the restrictions at the time – may have excluded people accessing higher levels of support, or those from more diverse backgrounds, especially those who are socially disadvantaged in some way. While caution is warranted when interpreting these findings and stress the importance of further investigation in a diverse Autistic sample, it is nevertheless encouraging that our findings corroborate those from Chan et al.[51], whose sample consisted mostly of Autistic men, whom they saw in person rather than online.

Second, one third of our sample reported a co-occurring diagnosis of attention deficit/hyperactivity disorder (ADHD). Including 'AuDHD' people should not be considered a limitation per se, since forms of neurodivergence often go hand-in-hand[80]. Nevertheless, it raises important questions about whether the nature of sociality, and in this case, weak-tie interactions, might be different for Autistic people with co-occurring ADHD compared to those without ADHD – and also to those ADHDers, who are not Autistic themselves. The social lives of ADHD people are under-researched, at least from a less pathologising lens[but see,81,82] and research is needed specifically on their weak-tie relationships. More work is needed to determine precisely how experiences of casual encounters are shared (or not) across neurodivergent profiles.

Finally, this study was not designed to examine directly Autistic people's peripheral relationships and, therefore, specific questions related to these relationships were not part of our interview schedule. Notwithstanding, a substantial amount of data spontaneously emerged on this topic, reflecting its apparent importance to people – young people and adults. The findings should provide an impetus for researchers to examine Autistic people's weak-tie interactions in greater depth, in partnership with Autistic people themselves.

## Conclusions

Our study demonstrates the potential power of weak ties for Autistic people. Breaking from the stereotypical assumption that Autistic people avoid social situations, with a proclivity to live in relative isolation, our study revealed that during the most intense periods of the COVID-19 pandemic, Autistic people missed not only their close personal relationships but a host of everyday interactions, with acquaintances and strangers. Moreover, they recognised that these weak-tie interactions were crucial to their wellbeing. Such a finding supports Fingerman's[11] thesis that "for individuals to flourish, to maximise their sense of connection and wellbeing, a wide variety of social ties may be imperative" (p. 69) and suggests that it applies to Autistic people as well as to non-autistic people. These findings should have important implications both for future research into Autistic sociality and for the design of practical services and supports to enhance Autistic people's opportunities to flourish.

## Data availability

Raw qualitative data cannot be made open access due to ethical restrictions. De-identified qualitative data are available from the corresponding author upon request.

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

## Acknowledgements

This project was funded through an Australian Research Council Future Fellowship, awarded to EP (FT190100077). The funder of the study had no role in study design, data collection, data analysis, data interpretation, or writing of this manuscript. We are extremely grateful to all our participants, who so generously gave up their time to convey their experiences during a very unsettling period. We also thank Simon Brett, Jac den Houting, Iliana

Magiati, Robyn Steward and Anna Urbanowicz, for their input into the original 'Everyday life during COVID-19' study, and to Marc Stears and Georgia Thomas-Parr for helpful comments on a previous draft of this manuscript.

## Author contributions

E.P.: Funding acquisition, Conceptualisation, Methodology, Investigation, Formal analysis, Writing – original draft. M.H.: Conceptualisation, Methodology, Investigation, Formal analysis, Writing – review and editing

## Competing interests

The authors declare no competing interests.
