## [Transparent Peer Review file · Communications Psychology]

Weak ties and the value of social connections for autistic people as revealed during the COVID-19 pandemic

Corresponding Author: Professor Elizabeth Pellicano

Version 0:

Decision Letter: first round

Dear Professor Pellicano,

Thank you for your patience during the peer-review process. Your manuscript titled "'I thought I was fine without people": Examining the power of weak ties for Autistic people during the COVID-19 pandemic" has now been seen by 2 reviewers, and I include their comments at the end of this message. They find your work of interest but raised some important points. We are interested in the possibility of publishing your study in Communications Psychology, but would like to consider your responses to these concerns and assess a revised manuscript before we make a final decision on publication.

We therefore invite you to revise and resubmit your manuscript, along with a point-by-point response to the reviewers. Please highlight all changes in the manuscript text file.

Editorially, we consider it important that the revised manuscript addresses the demographic limitations as noted by Reviewer 1 and the possible contribution of ADHD as noted by Reviewer 2.

I am attaching an Editorial Requests Table that details critical reporting requirements for the revised manuscript. Please attend to each item and ensure your manuscript is fully compliant. We are requesting that your manuscript aligns with these requirements as this facilitates the evaluation of your manuscript, reducing delays in re-review and potential future acceptance. If your revised manuscript is not aligned with these requests on major issues, such as those concerning statistics, it may be returned to you for further revisions without re-review. Additional information can be found in our style and formatting guide Communications Psychology formatting guide.

Please use the following link to submit your

- revised manuscript,
- point-by-point response to the referees' comments,
- cover letter (as a separate document),
- the Editorial Policy Checklist (see below),
- the Reporting Summary (see below), and
- the completed Editorial Request Table (attached):

Link Redacted

We hope to receive your revised paper within 8 weeks; please let us know if you aren't able to submit it within this time so that we can discuss how best to proceed. If we don't hear from you, and the revision process takes significantly longer, we may close your file. In this event, we will still be happy to reconsider your paper at a later date, provided it still presents a

significant contribution to the literature at that stage.

Best regards,

Jennifer Bellingtier

Jennifer Bellingtier, PhD
Senior Editor
Communications Psychology

REVIEWER EXPERTISE:

Reviewer #1 Autism, Qualitative Research

Reviewer #2 Autism, Qualitative Research

REVIEWER REPORTS:

Reviewer #1 (Remarks to the Author):

Thank you for the opportunity to review this manuscript about the power of weak ties for Autistic people as evidenced by experiences during the stay-at-home orders of the COVID-19 pandemic. I believe this paper will make a substantial contribution to the literature by illustrating the importance of “weak tie” social relationships for Autistic people and the desire Autistic people have for interpersonal connection. Further, your analytic methods are well and thoroughly described. I do have some suggestions to improve the paper before publication.

On p. 3, lines 37-38, you cite the DSM-5 to support the prior focus on Autistic people’s close ties only. Could you please clarify where in the DSM-5 this appears?

In the first full paragraph on p. 4, you move rather quickly from Granovetter’s 1973 work to Rajkumar et al.’s 2022 work on weak ties. It might help to contextualize your findings from 2020 if you say more about Rajkumar et al. (2022), given that ~50 years passed between the works you cite here.

In Table 1, please check the superscript characters that direct readers to the table note. I think the c next to “Man/boy” should be b, and the d next to “Co-occurring conditions” should be c.

Please clarify why young people’s parents, rather than the young people themselves, completed the online questionnaire. Was this true for all young people, or just some? I understand the justification for obtaining informed consent from parents of minors, but the potential for disagreement between self-report and parent-report makes me question its utility for collecting demographic, diagnostic, and background details.

Please add a subheading in the “Data analysis” section when you begin talking about researcher positionality.

Please name your main themes in the first paragraph of the Results.

I’m torn about your use of direct quotations to name themes and subthemes. On the one hand, it suggests authenticity of representation of your participants’ views. On the other hand, it suggests some lack of interpretation on the part of the researchers. Please consider how you have balanced these two needs and whether providing some more interpretation of your themes beyond choosing exemplary quotations might improve the quality of the work.

In your discussion of subtheme 1.3 about missing casual, incidental interactions, I wonder how much of it has to do with the social aspect and how much has to do with routines being disrupted. This appears again in your discussion of subtheme 3.1, where 203Adult mentions using “the same script that you just run every time.” I think that adding some content about the value of established routines for Autistic people could enrich your Discussion.

I would like to see more detailed discussion of the sample demographics in your Limitations. Specifically, please address the incredibly high number of women relative to men in your sample and the prevalence of adult diagnosis. It would also be helpful to discuss your demographics (including high education) in the context of recent publications about sampling bias in autism research (e.g., Rødgaard et al., 2022, *Autism Research*).

There are a few places in the manuscript where changing figurative language to literal language might help your readers.
1) p. 5, line 95: instead of “cognitive building blocks,” consider saying “cognitive skills.” If “building blocks” is the original language used by Baron-Cohen (1995), please put it in quotation marks.

2) The title of Theme 3: "Incidental human contact fills up my social cup." This could be interpreted either as fulfilling the person's social needs, or as overwhelming them with social contact so they run out of social resources/bandwidth. Literal language would help to resolve this ambiguity.

Please check carefully that all quotations have closing quotation marks and proper attribution.

Please check carefully that all instances of "casual" are spelled correctly and not substituted with "causal."

Please carefully consider your use of the word "neurotypical" and whether it would be more appropriate to say "non-autistic" or "allistic." There are many ways to be neurodivergent without being Autistic.

I also want to compliment specific sections of this work:

- 1) The first paragraph on p. 6 about recent critiques of the narrow focus on Autistic social challenges—really well done!
- 2) The first full paragraph on p. 26 about the likelihood that Autistic people's weak-tie interactions are still different from non-autistic people's, and your emphasis on the importance of choice and control for Autistic people—very important! I hope that future researchers and practitioners will take this to heart.

Thank you again for the opportunity to review this work. I look forward to reading its next iteration.

Reviewer #2 (Remarks to the Author):

Many thanks for the opportunity to read your interesting paper! It was an exciting read!

I mainly have two observations, or two things which may contribute to strengthen the theoretical contribution of the (already) strong paper.

1) Autistic vs ADHD sociality

As a third of your participants are Autistic ADHDees (combined autism and ADHD) I think there is a need to address this in the discussion – possible impact not only of Autistic but also of ADHD forms of sociality. I am mindful here that ADHD forms of sociality from a neurodiversity/neuroaffirmative approach is severely underresearched. So this may be both a suggestion for some more sentences in the conclusions in this paper and another paper (where you could dig some deeper into possible differences between your "only Autistic" and "AuDHD" sample).

2) weak ties- focused interest/interest-based sociality

You refer to work by Bertilsdotter Rosqvist 2019 on Autistic sociality, but I think it could be more developed - as Bertilsdotter Rosqvist is writing on Autistic sociality as both contextual and interest-based. So a possible development in the discussion could be a bit more on impact on possibility to engage in focused interests/passions – what is referred to here as "big Autie passions" (but could also be work/co-workers if one have ones focused interest as work) or "interest-focused social interactions" - on well being. So not directly connected to weak ties, but consequences of not being able to do weak ties-socialising on one's ability to perform ones focused interests (performing ones focused interests as strongly connected to ones wellbeing as an Autistic person)

EDITORIAL POLICIES

We ask that you ensure your manuscript complies with our editorial policies and reporting requirements.

To that end, we require revised manuscripts to be accompanied by two completed items: a reporting summary that collects information on study design and procedure, and an editorial policy checklist that verifies compliance with all required editorial policies.

- <https://www.nature.com/documents/nr-reporting-summary.zip>>Nature Research Reporting Summary
- <https://www.nature.com/documents/nr-editorial-policy-checklist.pdf>>Editorial Policy Checklist

All points on the policy checklist must be addressed. Your revised manuscript can only be sent back to the referees if these checklists are completed and uploaded with the revision.

Notes: If you have submitted a Stage 1 Registered Report, Review, Primer, Comment, or Perspective you do not need to submit these forms. If you have already submitted these forms, you may disregard this request.

Version 1:

Decision Letter: second round

Dear Professor Pellicano,

Your manuscript titled "“I thought I was fine without people”: Examining the power of weak ties for Autistic people during the COVID-19 pandemic" has now been seen by our reviewers, whose comments appear below. In light of their advice I am delighted to say that we are happy, in principle, to publish a suitably revised version in Communications Psychology.

We therefore invite you to revise your paper one last time to address the remaining concerns of our reviewers and a list of editorial requests. At the same time we ask that you edit your manuscript to comply with our format requirements and to maximise the accessibility and therefore the impact of your work.

EDITORIAL REQUESTS:

SUBMISSION INFORMATION:

OPEN ACCESS:

* **TRANSPARENT PEER REVIEW:** Communications Psychology uses a transparent peer review system. On author request, confidential information and data can be removed from the published reviewer reports and rebuttal letters prior to publication. If you are concerned about the release of confidential data, please let us know specifically what information you

would like to have removed. Please note that we cannot incorporate redactions for any other reasons.

* **DATA AVAILABILITY:**

Link Redacted

Best regards,

Jennifer Bellingtier

Jennifer Bellingtier, PhD
Senior Editor
Communications Psychology

REVIEWERS' EXPERTISE:

Reviewer #1 Autism, Qualitative Research
Reviewer #2 Autism, Qualitative Research

REVIEWERS' COMMENTS:

Reviewer #1 (Remarks to the Author):

Thank you for the opportunity to review this revision of "I thought I was fine without people." My prior comments have been sufficiently addressed.

I still think that the Introduction would benefit from more detail about the participants and findings of the Rajkumar et al. (2022) study, similar to how the authors describe Granovetter's "pioneering study of Boston-based workers" in which 84% found out about their jobs from casual contacts. However, I do not think the absence of these details about the Rajkumar et al. (2022) study should preclude acceptance for publication.

Reviewer #2 (Remarks to the Author):

Many thanks for the opportunity to read the revised version of your MS. I find the revisions made very satisfying and look forward seeing it published in the journal.

Response to reviewers

Reviewer 1

1. *“Thank you for the opportunity to review this manuscript about the power of weak ties for Autistic people as evidenced by experiences during the stay-at-home orders of the COVID-19 pandemic. I believe this paper will make a substantial contribution to the literature by illustrating the importance of “weak tie” social relationships for Autistic people and the desire Autistic people have for interpersonal connection. Further, your analytic methods are well and thoroughly described. I do have some suggestions to improve the paper before publication.”*

Response: Thank you – we are very grateful for the reviewer’s positive comments on our manuscript, and for their constructive feedback.

2. *“On p. 3, lines 37-38, you cite the DSM-5 to support the prior focus on Autistic people’s close ties only. Could you please clarify where in the DSM-5 this appears?”*

Response: Thank you. We were referring to criterion A3 in the DSM-5, which states that one key part of being Autistic is “deficits in developing, maintaining, and understanding relationships”, especially with peers – a feature that has persisted in the DSM since DSM-III (APA, 1980). That said, we appreciate that this is not straightforward to explain efficiently so have removed the reference from this part of the text.

3. *“In the first full paragraph on p. 4, you move rather quickly from Granovetter’s 1973 work to Rajkumar et al.’s 2022 work on weak ties. It might help to contextualize your findings from 2020 if you say more about Rajkumar et al. (2022), given that ~50 years passed between the works you cite here.”*

Response: Thank you. We now acknowledge explicitly the passage of time in this regard (see p. 4), while also noting that the purpose of the paper is not to conduct a full literature review on weak ties.

4. *“In Table 1, please check the superscript characters that direct readers to the table note. I think the c next to “Man/ boy” should be b, and the d next to “Co-occurring conditions” should be c.”*

Response: Thank you for spotting these errors! We have now corrected the superscript characters in Table 1.

5. *“Please clarify why young people’s parents, rather than the young people themselves, completed the online questionnaire. Was this true for all young people, or just some? I understand the justification for obtaining informed consent from parents of minors, but the potential for disagreement between self-report and parent-report makes me question its utility for collecting demographic, diagnostic, and background details.”*

Response: This is an important point. We should have clarified in the text that all young people, but especially those aged 16 years and over who consented for themselves, were given the choice to complete the background questionnaire or for their parent to do it on their behalf. Only one of the young people (aged 18 years) agreed to complete the online questionnaire themselves. We have now made this clearer in the text (see p. 11). We appreciate the issue the reviewer raises here but, as a team, we were highly cognisant of asking too much from people, especially Autistic young people, during an extraordinarily difficult time, which is why young people were given the choice.

6. *“Please add a subheading in the “Data analysis” section when you begin talking about researcher positionality.*

Response: We have now added this subheading (see p. 12).

7. *“Please name your main themes in the first paragraph of the Results.”*

Response: Our main themes now appear in the first paragraph of the Results section (see pp. 12-13).

8. *“I’m torn about your use of direct quotations to name themes and subthemes. On the one hand, it suggests authenticity of representation of your participants’ views. On the other hand, it suggests some lack of interpretation on*

the part of the researchers. Please consider how you have balanced these two needs and whether providing some more interpretation of your themes beyond choosing exemplary quotations might improve the quality of the work.”

Response: Thank you for raising this issue. We prefer to use participant quotes for our theme and subtheme labels. We are clear in our data analysis section that we analysed the data at a semantic (rather than a latent) level, and therefore ‘staying close’ to participants’ language. In our view, this does not indicate a lack of interpretation on our part as the *choice* of quotation is dependent on our interpretative analysis and reflects the arguments that we are presenting.

9. “In your discussion of subtheme 1.3 about missing casual, incidental interactions, I wonder how much of it has to do with the social aspect and how much has to do with routines being disrupted. This appears again in your discussion of subtheme 3.1, where 203Adult mentions using “the same script that you just run every time.” I think that adding some content about the value of established routines for Autistic people could enrich your Discussion.”

Response: We acknowledge this point, and the reviewer is clearly right to point out that disruption to routine *per se* is of concern to many Autistic people. Nonetheless, our purpose in the paper here was to emphasise the role of the disruption to a *particular* kind of routine – those which include casual social interactions, including with those who are otherwise strangers. As requested, we now raise this issue on p. 24.

10. “I would like to see more detailed discussion of the sample demographics in your Limitations. Specifically, please address the incredibly high number of women relative to men in your sample and the prevalence of adult diagnosis. It would also be helpful to discuss your demographics (including high education) in the context of recent publications about sampling bias in autism research (e.g., Rødgaard et al., 2022, *Autism Research*).”

Response: Many thanks for highlighting this issue. We agree that addressing sampling biases in autism research – and in psychological research more broadly – are much needed. We have since provided a more detailed discussion of the limitations of the characteristics of our sample (see pp. 27-28).

11. “There are a few places in the manuscript where changing figurative language to literal language might help your readers.
1) p. 5, line 95: instead of “cognitive building blocks,” consider saying “cognitive skills.” If “building blocks” is the original language used by Baron-Cohen (1995), please put it in quotation marks.
2) The title of Theme 3: “Incidental human contact fills up my social cup.” This could be interpreted either as fulfilling the person’s social needs, or as overwhelming them with social contact so they run out of social resources/bandwidth. Literal language would help to resolve this ambiguity.”

Response: This is a good point – thank you. We have changed the phrase “cognitive building blocks” to “cognitive skills” (see p. 5), as suggested, and we have also changed the title of Theme 3 (see p. 18). We have also double-checked the manuscript for places where we could improve its accessibility.

12. “Please check carefully that all quotations have closing quotation marks and proper attribution.”

Response: Done – thank you. Note that our longer (40+ words), indented quotes do not have quotation marks, as per APA format.

13. “Please check carefully that all instances of “casual” are spelled correctly and not substituted with “causal.”

Response: Apologies for this error – and thank you for picking it up. We have corrected these in the text.

14. “Please carefully consider your use of the word “neurotypical” and whether it would be more appropriate to say “non-autistic” or “allistic.” There are many ways to be neurodivergent without being Autistic.”

Response: We very much agree with your comment and have changed the word “neurotypical” to “non-autistic”, where appropriate.

15. “I also want to compliment specific sections of this work:
1) The first paragraph on p. 6 about recent critiques of the narrow focus on Autistic social challenges—really well

done!

2) *The first full paragraph on p. 26 about the likelihood that Autistic people's weak-tie interactions are still different from non-autistic people's, and your emphasis on the importance of choice and control for Autistic people—very important! I hope that future researchers and practitioners will take this to heart.*

Thank you again for the opportunity to review this work. I look forward to reading its next iteration.”

Response: Once again, thank you for your kind words – and especially for your thorough comments, which have very much strengthened our manuscript.

Reviewer 2

16. *“Many thanks for the opportunity to read your interesting paper! It was an exciting read!*

I mainly have two observations, or two things which may contribute to strengthen the theoretical contribution of the (already) strong paper.”

Response: We are very grateful for the reviewer for their encouraging comments, and for both observations, which we agree are important issues to include and discuss.

17. *“(1) Autistic vs ADHD sociality*

As a third of your participants are Autistic ADHDers (combined autism and ADHD) I think there is a need to address this in the discussion – possible impact not only of Autistic but also of ADHD forms of sociality. I am mindful here that ADHD forms of sociality from a neurodiversity/ neuroaffirmative approach is severely underresearched. So this may be both a suggestion for some more sentences in the conclusions in this paper and another paper (where you could dig some deeper into possible differences between your “only Autistic” and “AuDHD” sample).”

Response: This is an important issue – which, since being Autistic and an ADHDer (AuDHD) is fairly common (Rong et al., 2021), is possibly an issue that affects a good deal of current research with Autistic adults. The possible influence of ADHD on Autistic people's weak-tie interactions is difficult to disentangle in the current, qualitative dataset, but it does, as the reviewer suggests, identify a significant issue for future research. We have since added a paragraph on this issue in the Discussion (see p. 28).

18. *“(2) weak ties- focused interest/interest-based sociality*

You refer to work by Bertilsdotter Rosqvist 2019 on Autistic sociality, but I think it could be more developed - as Bertilsdotter Rosqvist is writing on Autistic sociality as both contextual and interest-based. So a possible development in the discussion could be a bit more on impact on possibility to engage in focused interests/passions – what is referred to here as “big Autie passions” (but could also be work/co-workers if one have ones focused interest as work) or “interest-focused social interactions” - on well being. So not directly connected to weak ties, but consequences of not being able to do weak ties-socialising on one's ability to perform ones focused interests (performing ones focused interests as strongly connected to ones wellbeing as an Autistic person)”.

Response: Thank you for encouraging us to include greater discussion on interest-based sociality and its implications, based on Hanna Bertilsdotter Rosqvist's work. We have since elaborated on this issue in the discussion (see pp. 23-24).

We thank the reviewers, once again, for their hugely constructive comments.

Response to reviewers

Reviewer 1

1. *"Thank you for the opportunity to review this revision of "I thought I was fine without people." My prior comments have been sufficiently addressed.*

I still think that the Introduction would benefit from more detail about the participants and findings of the Rajkumar et al. (2022) study, similar to how the authors describe Granovetter's "pioneering study of Boston-based workers" in which 84% found out about their jobs from casual contacts. However, I do not think the absence of these details about the Rajkumar et al. (2022) study should preclude acceptance for publication."

Response: We thank the reviewer for their positive comments on our revisions. As suggested, we have elaborated on the findings from the Rajkumar et al. (2022) study in the introduction (see p. 4).

Reviewer 2

2. *"Many thanks for the opportunity to read the revised version of your MS. I find the revisions made very satisfying and look forward seeing it published in the journal."*

Response: Thank you. Once again, we are extremely grateful for the encouraging comments on our manuscript.